# Type IV Pilus Shapes a 'Bubble-Burst' Pattern Opposing Spatial Intermixing of Two Interacting Bacterial Populations

Miaoxiao Wang,[a,b,c,d] Xiaoli Chen,[a,e] Yinyin Ma,[b,c] Yue-Qin Tang,[d] David R. Johnson,[c] Yong Nie,[a] Xiao-Lei Wu[a,e,f]

aCollege of Engineering, Peking University, Beijing, China
bDepartment of Environmental Systems Science, ETH Zürich, Zürich, Switzerland
cDepartment of Environmental Microbiology, Eawag, Dübendorf, Switzerland
dCollege of Architecture and Environment, Sichuan University, Chengdu, China
eInstitute of Ocean Research, Peking University, Beijing, China
fInstitute of Ecology, Peking University, Beijing, China

**ABSTRACT** Microbes are social organisms that commonly live in sessile biofilms. Spatial patterns of populations within biofilms can be important determinants of community-level properties. Spatial intermixing emerging from microbial interaction is one of the best-studied characteristics of spatial patterns. The specific levels of spatial intermixing critically contribute to how the dynamics and functioning of such communities are governed. However, the precise factors that determine spatial patterns and intermixing remain unclear. Here, we investigated the spatial patterning and intermixing of an engineered synthetic consortium composed of two mutualistic *Pseudomonas stutzeri* strains that degrade salicylate via metabolic cross-feeding. We found that the consortium self-organizes across space to form a previously unreported spatial pattern (here referred to as a 'bubble-burst' pattern) that exhibits a low level of intermixing. Interestingly, when the genes encoding type IV pili were deleted from both strains, a highly intermixed spatial pattern developed and increased the productivity of the entire community. The intermixed pattern was maintained in a robust manner across a wide range of initial ratios between the two strains. Our findings show that the type IV pilus plays a role in mitigating spatial intermixing of different populations in surface-attached microbial communities, with consequences for governing community-level properties. These insights provide tangible clues for the engineering of synthetic microbial systems that perform highly in spatially structured environments.

**IMPORTANCE** When growing on surfaces, multispecies microbial communities form biofilms that exhibit intriguing spatial patterns. These patterns can significantly affect the overall properties of the community, enabling otherwise impermissible metabolic functions to occur as well as driving the evolutionary and ecological processes acting on communities. The development of these patterns is affected by several drivers, including cell-cell interactions, nutrient levels, density of founding cells, and surface properties. The type IV pilus is commonly found to mediate surface-associated behaviors of microorganisms, but its role on pattern formation within microbial communities is unclear. Here, we report that in a cross-feeding consortium, the type IV pilus affects the spatial intermixing of interacting populations involved in pattern formation and ultimately influences overall community productivity and robustness. This novel insight assists our understanding of the ecological processes of surface-attached microbial communities and suggests a potential strategy for engineering high-performance synthetic microbial communities.

**KEYWORDS** biofilms, metabolic cross-feeding, spatial intermixing, spatial patterns, type IV pilus

Address correspondence to Yong Nie, nieyong@pku.edu.cn, or Xiao-Lei Wu, xiaolei_wu@pku.edu.cn.

The authors declare no conflict of interest.

In addition to the planktonic lifestyle, microorganisms also form intricate multispecies communities on surfaces (1, 2). These surface-attached communities play important roles in ecosystem processes (3), pollutant removal (4), and human health (5). Biofilms are spatially well-organized, with different populations interacting with each other, arranging themselves nonrandomly across space, and ultimately developing well-organized spatial patterns (here referred to as 'spatial self-organization') (6, 7). Spatial patterns of a community reflect the distribution of different populations across the habitat, and this distribution profoundly influences the interactions that occur among these populations. For example, spatial mixing of different interacting cells (here referred to as a 'mixed pattern') benefits their metabolic exchanges (8) and alleviates antibiotic stress (9). On the other hand, spatial demixing (here referred to as a 'segregated pattern') can stabilize intransitive interactions among antibiotic-producing, -sensitive, and -resistant species (10), and also protects cells from contact-dependent killing by their competitors (11). These outcomes can eventually be magnified to determine community-level properties, such as overall community productivity (8), resistance to invaders (12, 13), and robustness to initial conditions (7, 14).

Recently, several abiotic and biotic factors have been reported to influence the intermixing level of spatial patterns. For instance, genetic drift at expanding frontiers during range expansion generally demixes populations during growth, resulting in a segregated pattern (15). However, the degree of intermixing can be increased by increasing the nutrient levels (16), increasing the density of founding cells (17), the presence of physical objects (18) and promoting cooperative metabolic interactions (19, 20). As a result, the community will self-organize into a more intermixed pattern (21). Nevertheless, whether there are other factors which impact the intermixing level of spatial patterns still remains to be elucidated.

Here, we explored a selection of factors potentially affecting spatial self-organization of a well-defined, two-strain consortium using the strains *Pseudomonas stutzeri* AN0010 and *P. stutzeri* AN0001. In our previous work, we found that that neither AN0010 nor AN0001 can grow alone using salicylate as the sole carbon source (22). When paired and in the presence of salicylate, the two strains grew and acted as a cross-feeding consortium: AN0010 degrades salicylate into the intermediate catechol, and AN0001 further degrades catechol to pyruvate and acetyl-CoA (Fig. 1A). These small molecules can be released into the medium to support the growth of strain AN0010 (22). Therefore, each strain depends on the other for survival and reproduction in the presence of salicylate, so that the two strains engage in a mutualistic interaction and act as a bilateral cross-feeding consortium in the presence of salicylate (Fig. 1A).

**A 'bubble-burst' pattern developed by the cross-feeding consortium.** To investigate the spatial self-organization of this community, we cultured the community on an agarose surface using salicylate as the sole carbon source. Based on findings from previous studies, we expected our cross-feeding community to self-organize into a highly mixed pattern (14, 19, 20, 23). Surprisingly, we instead observed a segregated pattern, where the cells of strain AN0010 formed bubble-like structures inside the colony with cells of strain AN0001 subsequently surrounding these bubbles (Fig. 1B; Fig. S1A in the supplemental material). During range expansion, cells of strain AN0010 expanded from these 'bubble' structures, similar to the bubble 'burst' into the expanding sectors. We therefore refer to this previously undescribed spatial pattern as the 'bubble-burst' pattern. Our analysis using three-dimensional confocal microscopy showed that cells of strain AN0010 assembled in the 'bubble' structure, exhibiting a 'bowl'-like geometrical morphology (Fig. S1B). Further analysis of the fluorescence intensities of the two genotypes showed that cells of strain AN0001 were mostly distributed around the bubble formed by strain AN0010 (Fig. S1C), suggesting that the metabolic interaction between the two populations still necessitated that the two populations be in close spatial proximity.

To quantitatively test whether the levels of mixing between the two populations within the 'bubble-burst' pattern were lower than those of previously reported

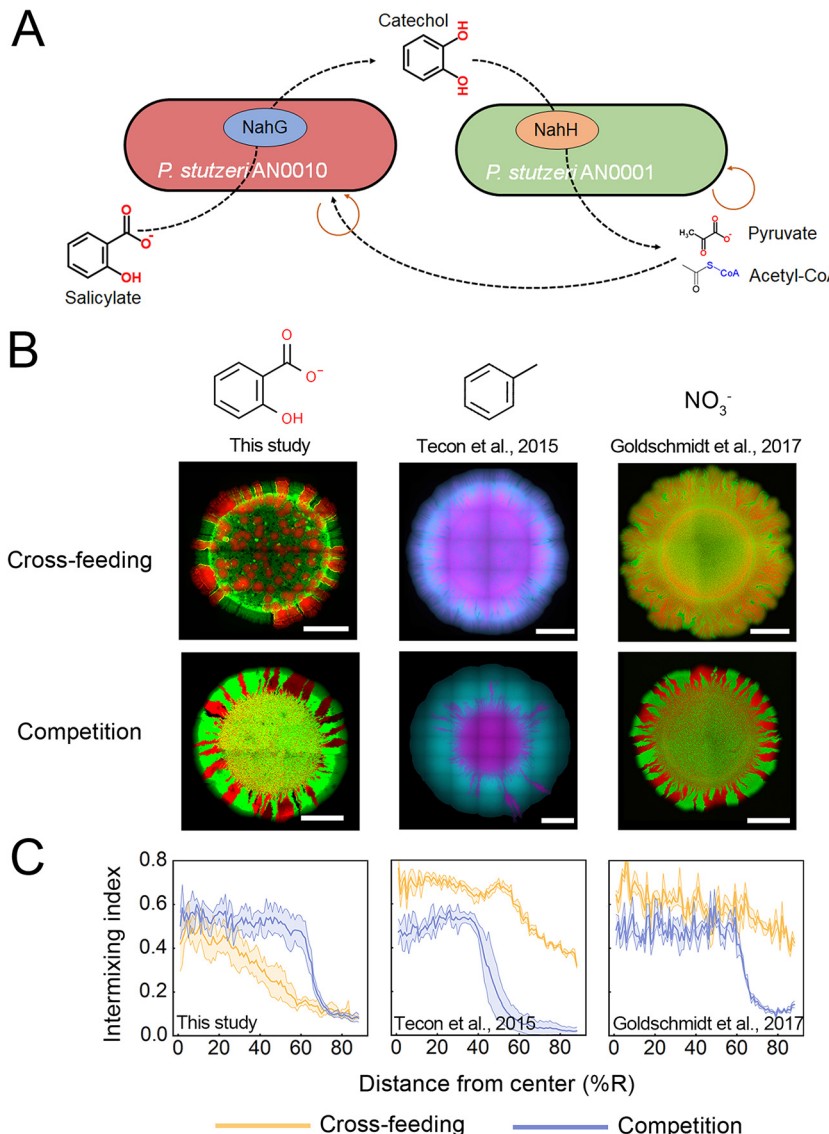

**FIG 1** The salicylate-degrading community self-organized into a 'bubble' burst pattern characterized by lower intermixing level. (A) Bilateral mutualistic interaction between strain *P. stutzeri* AN0010 and strain *P. stutzeri* AN0001 during salicylate degradation. Strain AN0010 degrades salicylate into the intermediate catechol, which feeds strain AN0001 as a substrate for further degradation. However, strain AN0010 cannot obtain a direct carbon source from salicylate degradation to support its growth. When AN0010 is paired with strain AN0001, AN0001 degrades catechol to pyruvate, feeding AN0010. (B) Representative colony patterns developed by our salicylate-degrading community, as well as those developed by similar, previously constructed cross-feeding communities. Overlay fluorescence images of these patterns are shown. For the patterns formed by our salicylate-degrading community (left), AN0010 was tagged with mCherry (shown as false-color red), while AN0001 was labeled with eGFP (shown as false-color green). Middle panels show the patterns developed by a toluene-degrading cross-feeding community built by Tecon et al. (14), in which strain *P. putida* PpF4 (tagged with eGFP, shown in cyan) degrades toluene to 3-methylcatechol, which is transformed to acetate and pyruvate by strain *P. putida* PpF107 (tagged with mCherry, shown in magenta). Right panels show the patterns developed by a denitrification cross-feeding community built by Goldschmidt et al. (23), in which strain *P. stutzeri* A1603 (tagged with eCFP, shown in green) converts $NO_3^-$ to $NO_2^-$, and strain *P. stutzeri* A1602 (tagged with mCherry, shown in red) further reduces $NO_2^-$ to $N_2$. Results of pattern formation assays in the 'Cross-feeding' (salicylate, toluene or $NO_3^-$ supplied in the medium, respectively) and 'Competition' scenarios (pyruvate, benzoate supplied as the sole carbon source, or two complete $NO_3^-$ degraders grown together on LB agar surface) are shown for all three synthetic communities. Scale bar in each image corresponds to 1 mm. (C) Analysis of intermixing indices of these patterns. Higher values indicate higher levels of local spatial intermixing of the two strains. These values were assessed through image analysis following a protocol modified from a previous study (23) (see Text S1 in the supplemental material for details). To obtain the patterns formed by our salicylate-degrading community, six experimental replicates were performed. Images of patterns formed by the toluene-degrading community were published (14) courtesy of Robin Tecon. In addition, microscopic images of the patterns formed by the denitrification community were obtained by performing three replicated pattern formation assays following a previously reported protocol (23).

patterns developed by other cross-feeding consortia, we calculated the intermixing indices of all these patterns (23) (see Text S1 in the supplemental material for details). We found that the intermixing levels of the 'bubble-burst' pattern were significantly lower than those generated by cross-feeding consortia performing toluene degradation (14) (unpaired, two-tailed Student's $t$ test: $P = 3.9e–12$) and denitrification (23) (Fig. 1C; Fig. S2; $P = 1.4e–11$). In addition, previous studies reported that spatial intermixing in those cross-feeding communities was reduced when the strains become metabolically independent (14, 19, 20, 23). We thus set out to test whether eliminating metabolic cross-feeding in our synthetic community also reduced the intermixing levels of the 'bubble-burst' pattern. We cultured our synthetic community on an agar surface using pyruvate as the sole carbon source. Pyruvate constitutes one of the final products of the salicylate degradation pathway and is directly utilized by both strains for growth (Fig. 1A). When the two strains directly competed for this limited resource, a clear segregated pattern formed (Fig. 1B). However, our analysis of this pattern suggested that the 'bubble-burst' pattern formed in the 'cross-feeding' scenario was even less mixed than the pattern formed in the 'competition' scenario (Fig. 1C; Fig. S2; $P = 2.7e–4$), especially at the inoculation area (Fig. 1C; Fig. S3), in disagreement with previous observations (14, 23) (Fig. 1B and C). In the 'cross-feeding' scenario, the average size of the sector formed by AN0010 was larger than that formed by AN0001 at the far radii, suggesting that this strain possesses higher fitness during range expansion in this scenario (Fig. S3). Moreover, we also tested whether the 'bubble-burst' pattern is observed regardless of the magnitude of known key drivers, for example, nutrient levels (16) and density of founder cells (17). These investigations indicated that although these factors quantitatively affected the detailed morphology of the pattern, alteration of these factors failed to totally disrupt the development of the 'bubble-burst' pattern (Fig. S4, S5).

**Removing the pili disrupts the 'bubble-burst' pattern and increases spatial intermixing.** It has previously been reported that cell appendages, such as type IV pilus and flagellum, are critically involved in the formation of *Pseudomonas* biofilms (24, 25). To investigate whether these cell appendages are also involved in the formation of our 'bubble-burst' pattern, we introduced loss-of-function deletions into genes encoding key proteins involved in pilus and flagellum assembly in both strains (Fig. S6). We found that the deactivation of flagellar genes did not change the development of the 'bubble-burst pattern'. In comparison, deactivation of the genes encoding type IV pilus caused the 'bubble-burst' pattern to disappear and significantly increased the spatial intermixing of the two interacting populations in the developed pattern (Fig. 2A and B; Fig. S2; $P = 1.9e–7$). The mixed pattern that formed also better resembled the ones which developed in previous, similar studies (14, 23) (Fig. 2A and B; Fig. S2; $P = 0.89$ compared with the pattern formed by the toluene-degrading community, and $P = 0.24$ compared with that of the denitrification community), and showed higher intermixing compared to the pattern formed by the same community in the 'competition' scenario (Fig. S2; $P = 1.1 e–8$). To test whether the disappearance of the 'bubble-burst' pattern only requires the pili mutation in a single strain, we examined the patterns formed by mixing the Δ*pili* mutant of one strain to the wild-type of the other strain. We found that the 'bubble-burst' pattern completely disappeared once the pili of strain AN0010 were knocked out (Fig. S7A). Moreover, the size of 'bubbles' significantly reduced when the *pili* mutant of AN0001 was mixed with the AN0010 strain (Fig. S7D and E). Together, these results strongly suggest that the presence of type IV pilus is a determining factor in the formation of the 'bubble-burst' pattern and controls the intermixing level of the spatial pattern.

**Removing the pili increased productivity and robustness of the surface-attached consortium.** To investigate whether increased intermixing following removal of type IV pili influences community-level properties, we compared the biomass of colonies developed by communities composed of wild-type strains and Δ*pili* mutant strains. Intriguingly, although the two communities grew similarly in liquid culture (22), colonies developed by the Δ*pili* mutants produced more biomass than those of the wild-type strains (Fig. 2C, unpaired two-tailed Student's $t$ test, $P = 0.011$; Fig. S7C). This

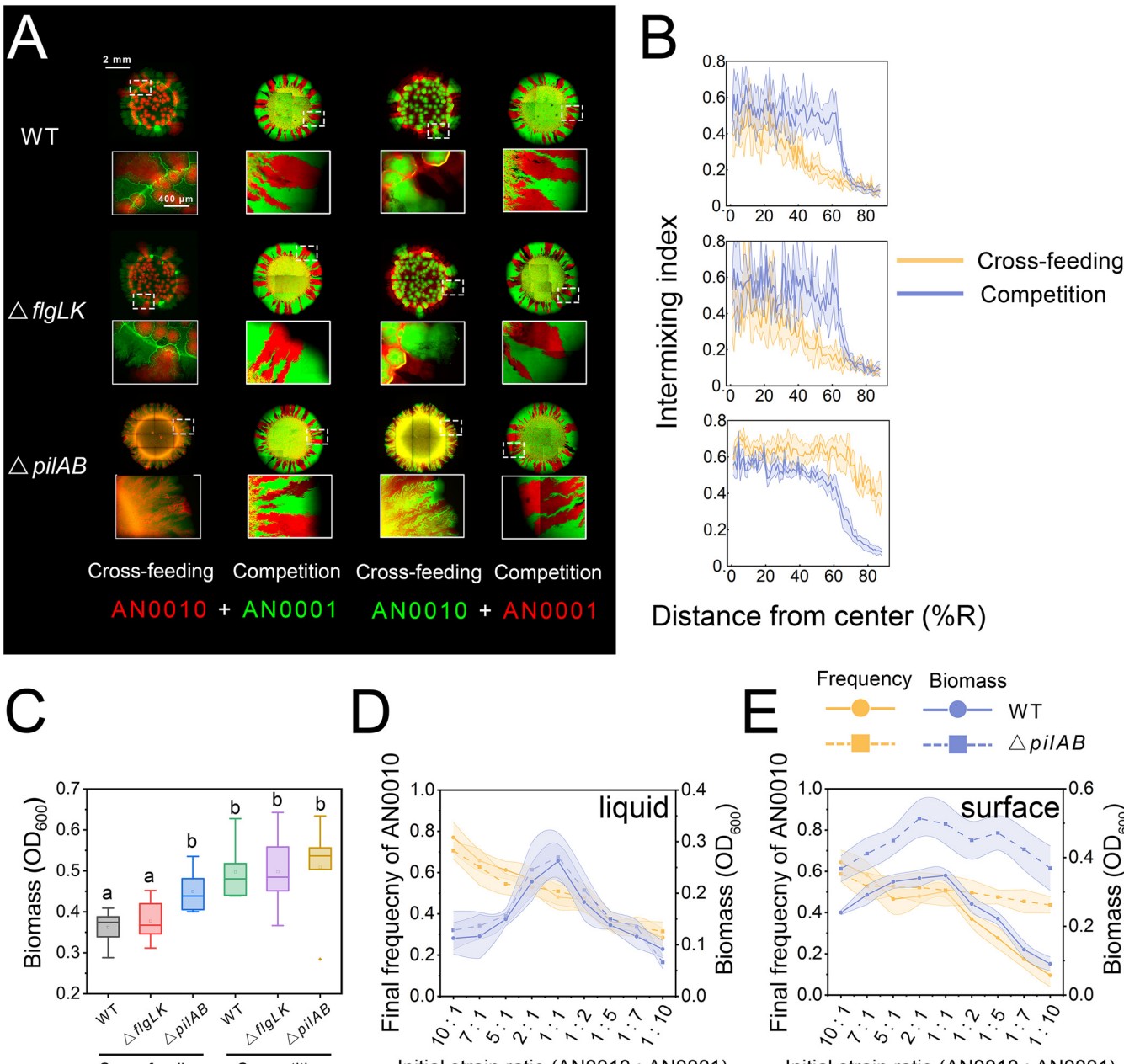

**FIG 2** Type IV pili are required for formation of 'bubble' structures, while flagella are dispensable. (A) Colony patterns formed by co-culturing wild-type strains of AN0010 and AN0001 (top), their flagellum-mutant strains (middle), and their pili-mutant strains (bottom) on agarose surface. Patterns obtained in both 'Cross-feeding' (supplying salicylate as the sole carbon source) and 'Competition' scenarios (supplying pyruvate as the sole carbon source) are shown. Alternative fluorescence labeling was used to eliminate potential effects caused by expression of different fluorescent proteins. Typical morphology of colony edges ($5\times$ zoom). Images were obtained after 120-h incubation. (B) Analyses of intermixing indices of these patterns. (C) Colony biomass ($OD_{600}$ [optical density at 600 nm]) analysis. Lowercase letters indicate significant differences between these conditions at $P = 0.05$ (unpaired, two-tailed Student's $t$ test). (D to E) Alterations of the salicylate-degrading community composed of wild-type (WT) strains (solid lines), as well as those of the community composed of the *pili* mutant (dashed lines), against the initial strain ratios between the two populations. We cultured these communities initiated with nine different strain ratios. After incubation (144 h for liquid cultivation, 120 h for cultivation on agarose surface), final community structures (yellow line) and biomass (blue line) were analyzed. In liquid cultivation, both communities exhibited similar alterations in community structures and biomass against different strain ratios (D). However, when the communities grew on agarose surface, for the community composed of *pili*-mutant strains, the final ratios of the two strains converged to similar values ($\approx$1:1) regardless of the initial strain ratio (E). In comparison, the final strain ratio for the community composed of wild-type strains exhibited larger variations. Furthermore, the initial strain ratio also showed smaller effects on the final productivity (biomass) of the *pili*-mutant community than on that of the wild-type community. These results strongly suggested that the *pili*-mutant community was more robust against fluctuations in initial conditions. For each initial strain ratio, six replicated experiments were performed. See Table S1 in the supplemental material for quantitative comparison of variations under different conditions.

result suggested that Δ*pili* mutants of the two strains interact better with each other than the wild-type strains, possibly due to spatial proximity to each other in the more mixed pattern, which leads to increased productivity at the community level. We then tested the robustness of community structure and productivity to the initial strain ratio, which was previously reported to constitute a key feature of the cross-feeding community (14, 19). Despite the fact that the two communities exhibited similar robustness to the initial strain ratio in liquid cultivation (Fig. 2D; see Table S1 for variation comparisons), we found that on agarose surface, the community composed of *pili* mutants was more robust to the initial conditions than the community that formed the 'bubble-burst pattern' (Fig. 2E; Table S1). Together, these results implied that removing the pili increased overall productivity, and the resulting community was more robust to variable initial ratios between the two strains.

## DISCUSSION AND CONCLUSION

How does type IV pilus contribute to the development of 'bubble-burst' pattern? Previous studies have indicated that type IV pilus plays essential roles in mediating twitching motility (26), stabilizing interactions between cells and the abiotic surface (24), and facilitating cell-to-cell adhesion (27, 28), which is required for aggregate formation. Several studies based on microscopic investigation directly observed that *Pseudomonas* cells harboring type IV pilus could migrate along the surface and gather together by recruiting adjacent and homologous cells, resulting in the formation of multicellular aggregates (24, 29). This pilus-mediated aggregating is affected by Psl exopolysaccharide (29), and the key genes involved in Psl synthesis are located in the genomes of our strains (Fig. S8). Therefore, we hypothesize that the observed 'bubble'-like structures in our study may be derived from pilus-mediated formation of cell aggregates. However, this hypothesis requires further testing on the single-cell level.

Here, we found that a cross-feeding community self-organized into a previously unknown 'bubble-burst' pattern in the presence of pili structures, which opposed spatial mixing of the different populations involved. Our findings also demonstrated that the reduced spatial mixing is associated with a decrease in community productivity and robustness. In conclusion, our findings suggest that the presence of cell constituents, such as type IV pili, may oppose metabolic interactions between different genotypes in surface-attached communities, up-scaling further to influence community-level properties. In addition, our results strongly suggest a potential strategy for engineering artificial communities with optimized spatial patterns: engineer the pili of the interacting strains to modulate interspecific distances in a surface-attached community, and we can thus promote its performance.

## SUPPLEMENTAL MATERIAL

Supplemental material is available online only.
**SUPPLEMENTAL FILE 1**, PDF file, 0.5 MB.

## ACKNOWLEDGMENTS

We thank Min Lin (Chinese Academy of Agricultural Sciences, Beijing, People's Republic of China) for providing plasmids pK18mobsacB and pRK2013, used for genetic engineering in this work; Ping Xu (Shanghai Jiao Tong University, Shanghai, People's Republic of China) for supplying plasmid pMMPc-Gm, used for fluorescence labeling in this study; Dani Or (ETH Zurich, Zurich, Switzerland) and Robin Tecon (NCCR Microbiomes and University of Lausanne) for providing the source images of their previous study; Martin Ackermann (ETH Zurich, Zurich, Switzerland) for constructive inputs on the design of this study; and T. Juelich (UCAS, Beijing) for linguistic assistance during the preparation of the manuscript.

This work was supported by the National Key R&D Program of China (2018YFA0902100 and 2018YFA0902103) and the National Natural Science Foundation of China (32130004, 91951204, 31770120, and 31770118).

We declare that we have no conflicts of interest.

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
