## [Reviewer comments · Microbiology Spectrum]

Microbiology Spectrum

Type IV pilus shapes a ‘bubble-jet’ pattern opposing spatial intermixing of two interacting bacterial populations

Miaoxiao Wang, Xiaoli Chen, Yinyin Ma, Yue-Qin Tang, David Johnson, Yong Nie, and Xiao-Lei Wu

Corresponding Author(s): Xiao-Lei Wu, College of Engineering, Peking University

Review Timeline:

Submission Date:	October 20, 2021
Editorial Decision:	November 22, 2021
Revision Received:	January 5, 2022
Editorial Decision:	January 23, 2022
Revision Received:	January 26, 2022
Accepted:	January 28, 2022

Editor: Benjamin Wolfe

Reviewer(s): The reviewers have opted to remain anonymous.

Transaction Report:

DOI: <https://doi.org/10.1128/Spectrum.01944-21>

November 22, 2021

Prof. Xiao-Lei Wu
College of Engineering, Peking University
Beijing
China

Re: Spectrum01944-21 (Type IV pilus shapes a 'bubble-jet' pattern opposing spatial intermixing of two interacting bacterial populations)

Dear Prof. Xiao-Lei Wu:

Thank you for submitting your manuscript to Microbiology Spectrum. Three experts have reviewed your work and have provided very helpful comments to make the manuscript stronger. Please carefully review these comments (at the bottom of this email) and use them to revise your manuscript.

Link Not Available

Sincerely,

Benjamin Wolfe

Journals Department
Reviewer comments:

Reviewer #1 (Comments for the Author):

Wang et al described how Type IV pilus can affect spatial patterning of a synthetic microbial community. The work is interesting.

Major comment:

It seems that the two strains engage in a commensal interaction? If so, then the work of Momeni et al predicts that the strains should not be intermixed. The strains should be segregated, with one possibly laying on top of the other. Regardless, authors should compare monoculture growth with coculture growth to establish whether the two strains engage in commensalism or mutualism.

Minor:

Line 142: check grammar.

Reviewer #2 (Comments for the Author):

The manuscript discusses the role of pili on spatial organization of synthetic communities consisting of two strains engaged in

cross-feeding. The authors show that the WT (with the pili) shows significantly lower spatial intermixing, compared to what is expected from cross-feeding. They confirm their observation by showing that more intermixing is observed with pili knock-out mutants.

Overall, the manuscript is well-written, clear, and easy to follow, in my opinion. The authors have examined the natural questions that arise and tested the corresponding cases. I do have a few comments and suggestions for the authors, as listed below.

Major comments:

1. The "cross-feeding" terminology is a little confusing because in the past it has been used both for unilateral interactions (e.g. A provides a metabolite for B) and bilateral interactions (e.g. A feeds B and B feeds A). From the design perspective of the introduced system in this manuscript, it seems the former type applies, but some of the arguments (including the expected intermixing) would apply to the latter type. I think this is a matter that the authors need to clarify. Specifically: (1) Is the expected interaction between the two strains unilateral or bilateral cross-feeding? (2) Is the expected interaction between the two strains commensalism (B benefits from A) or mutualism (B benefits from A and A benefits from B).

Minor comments:

1. The authors have mentioned their investigations with pili knockouts of one strain at a time affects intermixing. In my opinion, this is an important finding and belongs in the main text, rather than the supplementary information.
2. I suggest moving the Materials and Methods to the main text, following the common template for Spectrum papers.
3. I suggest dividing the Results section into subsections with subheadings that highlight each finding. In my opinion, this would improve the narrative of the paper and better highlight the authors' findings.
4. In Fig S5D-E, I suggest using the explicit labels for the x-axis, rather than the designations "1" and "2".

Reviewer #3 (Comments for the Author):

This manuscript examines spatial pattern formation in two *Pseudomonas stutzeri* strains which have been engineered to be obligate cross-feeders when growing on salicylate. Previous studies in other systems showed that crossfeeding pairs generate an intermixed pattern during range expansion, and the authors wondered whether the same was true for their crossfeeding community, and furthermore what the effect of type IV pili and flagella were on the observed pattern. They used microscopy to show that in the inoculation area, an unusual pattern formed (deemed "bubbles"), and beyond the inoculation area in range expansion, a form of sectoring occurred ("jet"). They analyzed the level of intermixing across the entire growth area with a simple, previously described metric (# of strain intersections divided by circumference being examined) and showed that by this metric, the crossfeeding community intermixed less than the competitive control, which contrasts with previous results in other crossfeeding systems. Deleting the pili, but not the flagella, abolished this "bubble-jet" phenomenon, and also increased total community productivity, hinting that a trade-off exists between maintaining pili and being optimized crossfeeders.

On the whole, this is an interesting study. It sheds light on how pattern formation is more nuanced than simply categorizing the ecological association between two strains (mutualist vs. competitors). However, there are some criticisms I would like to see addressed.

One criticism is that the organization of the manuscript lead me to confusion a few times. The main question being asked is whether the presence of pili distort the intermixing pattern observed by others, because pili are known to be involved in aggregation (which may prevent intermixing). It is then confusing on lines 97-98 to hear the hypothesis that the WT strains should intermix similar to other strains, given that they have pili. Then, a large chunk of the manuscript is dedicated to characterizing the pattern in WT cells (which have pili) before getting to the main experiment in the paper (deleting the pili and reassessing the pattern). I would argue that the manuscript would be much clearer if Fig. 1 showed the results of the main test: the patterning of crossfeeders in the presence vs. absence of the pili. This would allow readers to see that the control result does occur when expected (i.e. there is a typical intermixing pattern when there are not pili), which is altered by the presence of pili. After making that main point, the further characterization could occur with a better flow.

Related to organization, there is a paragraph near the end which introduces the aggregative phenotype which T4 pili cause. This seems like introductory material, as it is one of the reasons why one might expect that crossfeeders with these pili may not be able to intermix so easily (because they get stuck together due to pili-mediated aggregation). In the current second paragraph, it isn't totally clear that the aggregation which T4 pili can cause is known to occur in these strains.

I have two main scientific criticisms. First, I am not convinced that the intermixing index used in this paper is capturing important nuance. The index is the number of intersections between the two populations divided by the circumference being analyzed. This does not take into account the size of each sector. A major claim in this paper is that the cross-feeding community is less intermixed than the competitive community, based upon this index. However, if I examine the microscopy in Fig. 1, I see that the green sectors are quite thin. Therefore, the green cells are usually much closer to the red cells in the cross-feeding community than is the case for the competitive community. I would argue that from the green cells' perspective, the cross-feeding

community is more intermixed beyond the inoculation area than is the competitive community. Even if one disagrees with that interpretation, in any case the patterns are quite clearly different, yet the index analysis says they are the same at far radii, meaning it is not capturing important information. Could a modified metric be created which captures the clear differences between the two colonies?

My second criticism is that the statistics are not well-enough explained. What specific values are being compared in the Mann-Whitney test? The intermixing indices at all radii, at just the range-expansion radii, ...? If all radii (or even more than one radius) were used, how was the non-independence of the datapoints controlled for (as one could do, for example, with a mixed-effects model)? Furthermore, it isn't clear to me that the interpretation is always valid. For example, on lines 148-149, those p-values suggest that the intermixing index is significantly different between the pili mutant crossfeeders and previous studies, yet the sentence says "better resembled the ones that developed in previous similar studies," which seems to be arguing for the absence of a significant difference. Please clarify.

Other:

The productivity results with the pili led the authors to an intriguing hypothesis: pili mutants interact better with each other than WT strains, possible due to the increased spatial proximity. Put differently, I believe this would lead the authors to hypothesize that the pili mutants have higher fitness than WT. It would be great to have a test of this, by doing an invasion assay of the pili mutant, and seeing if it increases in frequency. Total community biomass, which is what was measured, may correlate to strain fitness, but they are not 1:1.

Line 135: it argues that changing nutrient concentration didn't affect the qualitative appearance of the bubble-jet pattern; please specify that you mean the average # of bubbles, as all other metrics (intermixing index, bubble area, total biomass, and I believe strain dominance) changed. Furthermore, there is no statistical support testing the # of bubbles, and when I look at S2E, it appears that the median number of bubbles increases monotonically until 10mmol, then stays at a plateau. I would argue this is contrary to the claim in the main text.

Abstract: should note the general reason why intermixing has been shown to occur (i.e. mutualism). Arguably one could claim that the best-studied spatial pattern isn't intermixing but sectoring of competitive / neutral populations.

Figure 1A is not clear for defining the cross-feeding relationship. Can AN0010 grow during its upstream partial degradation? Or does it require pyruvate / acetyl-CoA to be generated by AN0001? A better description of the specific needs of each partner is needed. I see this is available in the supp, but I think it is critical enough to be in the main text or Fig 1. I also think proof that they cannot grow on their own on sallycilate is important.

Line 68: a bit too strong to say that spatial mixing alleviates antibiotic stress-perhaps add "can," as it is only referencing one instance

line 76: "genetic surfing" is jargon, please define or rewrite

line 80: "metabolic interactions" is unclear; please specify mutualistic / cooperative

Line 142: sentence fragment

Line 178: grammar mistake

Fig 2D, E: I cannot distinguish the circles and squares. I would argue final frequency and biomass should be separate graphs.

Lines 100-101: it argues that AN0010 formed bubble-like structures, then AN0001 surrounded these, but I see no temporal analysis besides a single time series in S1 to support this claim.

"jetted" to me implies movement with force; I think it will lead to confusion and the authors should choose a different descriptor.

Staff Comments:

Preparing Revision Guidelines

To submit your modified manuscript, log onto the eJP submission site at <https://spectrum.msubmit.net/cgi-bin/main.plex>. Go to Author Tasks and click the appropriate manuscript title to begin the revision process. The information that you entered when you first submitted the paper will be displayed. Please update the information as necessary. Here are a few examples of required

updates that authors must address:

Please return the manuscript within 60 days; if you cannot complete the modification within this time period, please contact me. If you do not wish to modify the manuscript and prefer to submit it to another journal, please notify me of your decision immediately so that the manuscript may be formally withdrawn from consideration by Microbiology Spectrum.

Responses to the Editor's comments

Thank you for submitting your manuscript to Microbiology Spectrum. Three experts have reviewed your work and have provided very helpful comments to make the manuscript stronger. Please carefully review these comments (at the bottom of this email) and use them to revise your manuscript.

Reponses: We sincerely thank you and the reviewers for thoroughly examining our manuscript and providing very helpful comments and suggestions to strength our manuscript. We have carefully studied these comments, and responded as fully as we could to address the concerns of the reviewers. In particular, we have (i) clarified that the two strains in our study show mutualistic interaction; (ii) revised Introduction and Discussion sections, as well as added subheadings, to improve the main flow of the manuscript; (iii) added more quantitative and statistical analyses to better solidify our conclusions. We hope that you will find this new version of our manuscript improved much.

Responses to the Reviewer's comments

Reviewer #1 (Comments for the Author):

Wang et al described how Type IV pilus can affect spatial patterning of a synthetic microbial community. The work is interesting.

Response: We greatly thank you for providing helpful comments to guide our revision. We have made great efforts to improve the manuscript following these suggestions. In particular, we clarify more rigorously that the two strains engage in mutualism but not commensalism in our synthetic system. **The page and line numbers in marked version of the revised manuscript (RM) where changes have been made were indicated in each response.** Hope these revisions can meet your approval.

Major comment:

It seems that the two strains engage in a commensal interaction? If so, then the work

of Momeni et al predicts that the strains should not be intermixed. The strains should be segregated, with one possibly laying on top of the other. Regardless, authors should compare monoculture growth with coculture growth to establish whether the two strains engage in commensalism or mutualism.

Response: We are sorry for not clarifying our synthetic system well. The two strains engage in a **mutualistic interaction**, which was experimentally verified in our previous study (1). Shown below is the data **adapted from our former study**,

Figure R1 Identification of the strains used in this study. (A) Growth dynamics of the mono-culture of four strains using salicylate (solid line) or catechol (dash line) as the sole carbon source. Five replicates were performed for each treatment. 10 C-mmol/L salicylate or catechol were used in the culture experiments. (B) Liquid co-culture dynamics of strain AN0010 and strain AN0001 supplying salicylate as the sole carbon source.

These results show that either strain AN0010 or strain AN0001 cannot grow alone using salicylate as the sole carbon source (Figure R1A); when paired and in presence of salicylate, the two strains grew and acted as a cross-feeding consortium (Figure R1B). In particular, strain AN0010 cannot grow even if it is able to degrade salicylate to catechol. We propose that this is because AN0010 cannot obtain direct carbon source from salicylate degradation to support its growth (1). Therefore, **both AN0010 and AN0001 benefit from the other, and each partner depends on the other for survival and reproduction, which is a typical association of obligate mutualism, as defined in the previous articles (2, 3)**. Accordingly, we expected that the strains should be intermixed when grow on a surface (4, 5).

To better clarify this mutualistic interaction, we made following revisions in the RM:

1. We clarified the that these two strains engage in a mutualistic interaction in **line 93-100**, as follow:

“Strain AN0010 degrades salicylate into the intermediate catechol, but cannot grow in monoculture using salicylate (26). Instead, when the two strains are paired, strain AN0001 further degrades catechol to pyruvate and acetyl-CoA (Figure 1A). These small molecules are released to the medium to support the growth of strain AN0010 (26). Therefore, these two strains engage in a mutualistic interaction, and act as a bilateral cross-feeding interaction consortium in the presence of salicylate (Figure 1A).”

2. The term ‘Mutualism’ is emphasized in Abstract in **line 29**.
3. We revised Figure 1A to clarify the mutualistic interaction between the two strains.

Figure 1A Schematic shows the bilateral mutualistic interaction between strain *P. stutzeri* AN0010 and strain *P. stutzeri* AN0001 during salicylate degradation. Strain AN0010 degrades salicylate into the intermediate catechol, which feeds strain AN0001 as the substrate for further degradation. However, strain AN0010 cannot obtain direct carbon source from salicylate degradation to support its growth. When paired with strain AN0001, AN0001 degrades catechol to pyruvate, which feeds AN0010 with pyruvate to support its growth.

Hope these explanations and revisions can address your concern.

Minor:

Line 142: check grammar.

Response: Sorry for the mistake. In the RM, the sentence has been modified (**line 153-154**), as follow:

“We found that the deactivation of flagellar genes did not change the development of the ‘bubble-burst pattern’.”

Reviewer #2 (Comments for the Author):

The manuscript discusses the role of pili on spatial organization of synthetic communities consisting of two strains engaged in cross-feeding. The authors show that the WT (with the pili) shows significantly lower spatial intermixing, compared to what is expected from cross-feeding. They confirm their observation by showing that more intermixing is observed with pili knock-out mutants.

Overall, the manuscript is well-written, clear, and easy to follow, in my opinion. The authors have examined the natural questions that arise and tested the corresponding cases. I do have a few comments and suggestions for the authors, as listed below.

Response: We greatly thank you for the positive evaluation of our paper, and insightful comments helping us improve the manuscript. In the revised manuscript (RM), we clarify that the two strains in our system show a bilateral cross-feeding interaction, and engage in mutualism. **The page and line numbers in marked version of the revised manuscript (RM) where changes have been made were indicated in each response.** Hope these revisions can meet your approval.

Major comments:

1. The "cross-feeding" terminology is a little confusing because in the past it has been used both for unilateral interactions (e.g. A provides a metabolite for B) and bilateral interactions (e.g. A feeds B and B feeds A). From the design perspective of the introduced system in this manuscript, it seems the former type applies, but some

of the arguments (including the expected intermixing) would apply to the latter type. I think this is a matter that the authors need to clarify. Specifically: (1) Is the expected interaction between the two strains unilateral or bilateral cross-feeding? (2) Is the expected interaction between the two strains commensalism (B benefits from A) or mutualism (B benefits from A and A benefits from B).

Response: We are sorry for this terminology confusion. In our study, the two strains show **bilateral mutualism**. Either the strain AN0010 or AN0001 cannot grow alone using salicylate as the sole carbon source (Figure R1A). Strain AN0010 can degrade salicylate into the intermediate catechol, but it cannot obtain direct carbon source from salicylate degradation to support its growth (1). While AN0001 cannot degrade salicylate. When these two strains are co-cultured together, the strain AN0010 feeds strain AN0001 catechol as the substrate and AN0001 degrades catechol to pyruvate, which is released to the medium. So AN0001 feeds AN0010 with pyruvate to support its growth. Therefore, the two strains show bilateral mutualism (as shown in Figure 1A in RM).

In our verification experiments (Figure R1; **adapted from our former study (1)**), both strain AN0010 and strain AN0001 cannot grow in monoculture using salicylate (Figure R1A). However, when paired and in presence of salicylate, the two strains significantly grew (Figure R1B). Therefore, **both AN0010 and AN0001 benefit from the other, and each partner depends on the other for survival and reproduction, which is a typical association of obligate mutualism, as defined in previous articles (2, 3).**

Figure R1 Identification of the strains used in this study. (A) Growth dynamics of the mono-culture of four strains using salicylate (solid line) or catechol (dash line) as the sole carbon source. Five replicates were performed for each treatment. 10 C-mmol/L salicylate or catechol were used in the culture experiments. (B) Liquid co-culture dynamics of strain AN0010 and strain AN0001 supplying salicylate as the sole carbon source.

To better clarify this terminology, we made following revisions in the RM:

1. We clarified the that these two strains engage in a mutualistic interaction in **line 93-100**, as follow:

“Strain AN0010 degrades salicylate into the intermediate catechol, but cannot grow in monoculture using salicylate (26). Instead, when the two strains are paired, strain AN0001 further degrades catechol to pyruvate and acetyl-CoA (Figure 1A). These small molecules are released to the medium to support the growth of strain AN0010 (26). Therefore, these two strains engage in a mutualistic interaction, and act as a bilateral cross-feeding interaction consortium in the presence of salicylate (Figure 1A).”

2. The term ‘Mutualism’ is emphasized in Abstract in **line 29**.
3. We revised Figure 1A to clarify the mutualistic and bilateral interaction between the two strains.

Figure 1A Schematic shows the bilateral mutualistic interaction between strain *P. stutzeri* AN0010 and strain *P. stutzeri* AN0001 during salicylate degradation. Strain AN0010 degrades salicylate into the intermediate catechol, which feeds strain AN0001 as the substrate for further degradation. However, strain AN0010 cannot obtain direct carbon

source from salicylate degradation to support its growth. When paired with strain AN0001, AN0001 degrades catechol to pyruvate, which feeds AN0010 with pyruvate to support its growth.

Hope these explanations and revisions can address your concern.

Minor comments:

1. The authors have mentioned their investigations with pili knockouts of one strain at a time affects intermixing. In my opinion, this is an important finding and belongs in the main text, rather than the supplementary information.

Response: Thank you for the suggestion. We agree that this is an important result. However, we wish to publish this manuscript as an 'Observation' type. The Word Count Guidance of the journal (<https://journals.asm.org/journal/spectrum/article-types>) suggests that the paper should be with a maximum of 2 figures. Although this result is vital, we think the two figures in the main text contains more important findings. Besides, the two figures are too large to be integrated with more information. Therefore, we suggest here we still keep this result in Supplementary Information. Hope this explanation can meet your approval.

2. I suggest moving the Materials and Methods to the main text, following the common template for Spectrum papers.

Response: Thank you for the suggestion. We tried to move Materials and Methods section to the Main Text. However, the initial-check editor gave us a suggestion that because this manuscript is submitted as an 'Observation' type, detailed description of methods is mandatory to be shown in Supplementary information. We are sorry about this setting but we think this may not bother the main flow of the manuscript. Hope this explanation can meet your approval.

3. I suggest dividing the Results section into subsections with subheadings that highlight each finding. In my opinion, this would improve the narrative of the paper

and better highlight the authors' findings.

Response: Thank you for this insightful suggestion. In the RM, we added several subheadings to highlight each finding, as follow:

- (1) “A ‘bubble-burst’ pattern developed by the cross-feeding consortium” (line 101 in RM)
- (2) “Removing the pili disrupts the ‘bubble-burst’ pattern and increases spatial intermixing” (line 146-147 in RM)
- (3) “Removing the pili increased productivity and robustness of the surface-attached consortium” (line 175-176 in RM)
- (4) “Discussion and conclusion” (line 196 in RM)

We believe main flow of RM becomes easier to follow.

4. In Fig S5D-E, I suggest using the explicit labels for the x-axis, rather than the designations "1" and "2".

Response: Thank you for the suggestion. The figure has been revised in the RM accordingly.

Reviewer #3 (Comments for the Author):

This manuscript examines spatial pattern formation in two *Pseudomonas stutzeri* strains which have been engineered to be obligate cross-feeders when growing on salicylate. Previous studies in other systems showed that crossfeeding pairs generate an intermixed pattern during range expansion, and the authors wondered whether the same was true for their crossfeeding community, and furthermore what the effect of type IV pili and flagella were on the observed pattern. They used microscopy to show that in the inoculation area, an unusual pattern formed (deemed "bubbles"), and beyond the inoculation area in range expansion, a form of sectoring occurred ("jet"). They analyzed the level of intermixing across the entire growth area with a simple, previously described metric (# of strain intersections divided by circumference being examined) and showed that by this metric, the crossfeeding community intermixed

less than the competitive control, which contrasts with previous results in other crossfeeding systems. Deleting the pili, but not the flagella, abolished this "bubble-jet" phenomenon, and also increased total community productivity, hinting that a trade-off exists between maintaining pili and being optimized crossfeeders.

On the whole, this is an interesting study. It sheds light on how pattern formation is more nuanced than simply categorizing the ecological association between two strains (mutualist vs. competitors). However, there are some criticisms I would like to see addressed.

Response: We sincerely thank the reviewer for the time and effort on evaluating our work. We have carefully revised the manuscript based on your valuable comments. In the revised manuscript (RM), we have (i) revised Introduction and Discussion section, as well as added subheadings, to improve the main flow of the manuscript; (ii) added more quantitative and statistical analyses to better solidify our conclusions; (iii) clarified that the two strains in our system show a bilateral cross-feeding interaction, and engage in mutualism. **The page and line numbers in marked version of the revised manuscript (RM) where changes have been made were indicated in each response.** Hope these revisions can meet your approval.

One criticism is that the organization of the manuscript lead me to confusion a few times. The main question being asked is whether the presence of pili distort the intermixing pattern observed by others, because pili are known to be involved in aggregation (which may prevent intermixing). It is then confusing on lines 97-98 to hear the hypothesis that the WT strains should intermix similar to other strains, given that they have pili. Then, a large chunk of the manuscript is dedicated to characterizing the pattern in WT cells (which have pili) before getting to the main experiment in the paper (deleting the pili and reassessing the pattern). I would argue that the manuscript would be much clearer if Fig. 1 showed the results of the main test: the patterning of cross-feeders in the presence vs. absence of the pili. This would allow readers to see that the control result does occur when expected (i.e. there is a

typical intermixing pattern when there are not pili), which is altered by the presence of pili. After making that main point, the further characterization could occur with a better flow.

Response: We are sorry for the confusing organization. We acknowledge that the question proposed in the old manuscript is confusing and deviates the main flow of the papers. Actually, our study is **phenomenon-driven** and does not start with asking the effects of pili. The main logic of the paper is that:

- (1) We ask whether there are other previously unknown factors that may contribute the intermixing of the patterns (type IV pilus is only one of the candidates).
- (2) We investigated this question by studying a pattern formed by a cross-feeding community. We observed a less intermixing ‘bubble-burst’ (‘bubble-jet’) pattern (**phenomenon**), which is opposed to previous observations and thus contrary to the expectation (**Figure 1**).
- (3) After test many potential factors, we found that **type IV pilus** is responsible for this less intermixing pattern, which is a factor uncharacterized previously (**Figure 2A-B**).
- (4) We further tested the **ecological significance** of our finding, and found that removing the pili increased productivity and robustness of the surface-attached consortium (**Figure 2C-E**).
- (5) We **discussed**/hypothesized the possible mechanism shape the ‘bubble-burst’ (‘bubble-jet’) pattern, and draw our **conclusion**.

We acknowledge that the pili-hypothesis in the introduction part, as well as the follow-up questioning, makes readers confusing. We thus moved these descriptions to the discussion part (see the response below), and re-write this part to clarify our starting point, as follow (line 89-90 in RM):

“Nevertheless, whether there are other factors that impact the intermixing level of the spatial patterns still remains to be elucidated.”

We also added four **subheadings to highlight each part** that we state above, as follow:

- (1) **Phenomenon section:** “A ‘bubble-burst’ pattern developed by the cross-feeding consortium” (line 101 in RM)
- (2) **Pili section:** “Removing pili disrupts the ‘bubble-burst’ pattern and increases spatial intermixing” (line 146-147 in RM)
- (3) **Ecological significance section:** “Removing the pili increased productivity and robustness of the surface-attached consortium” (line 175-176 in RM)
- (4) “Discussion and conclusion” (line 196 in RM)

We believe main flow of RM becomes easier to follow.

Related to organization, there is a paragraph near the end which introduces the aggregative phenotype which T4 pili cause. This seems like introductory material, as it is one of the reasons why one might expect that cross-feeders with these pili may not be able to intermix so easily (because they get stuck together due to pili-mediated aggregation). In the current second paragraph, it isn't totally clear that the aggregation which T4 pili can cause is known to occur in these strains.

Response: Continuing the above response, we are sorry for this confusing discussion. In the RM, we do not hypothesize that ‘the presence of pili distorts the intermixing pattern’ at the beginning (as we state in the last response). Instead, we found that type IV pili distort the intermixing pattern after our experiments. **The goal of this discussion paragraph is to discuss the potential mechanisms behind our finding (that is, why the presence of pili shapes the ‘bubble-burst’ (‘bubble-jet’) pattern). After discussion, we proposed our hypothesis that needs to be further tested.**

In order to investigate whether the pili-mediated aggregation is known to occur in our strains, we need technology of quantitative single-cell measurement of microorganisms to directly track the formation of such aggregates. We apologize that we have not established such a platform in our lab at this point. We hope, in the future, we can shed light on the underlying mechanism of how the ‘bubble-burst’ (‘bubble-jet’) pattern was self-organized. However, we think these studies are a little bit beyond the goal of this paper, which mostly aimed to prove that Type IV pilus is important to affect intermixing level of the pattern. Nevertheless, we can provide an indirect

experimental evidence at genome level. Our strains possess complete Type IV pilus gene clusters, and the key genes encoding Psl exopolysaccharide, which are very similar as well-characterized Psl producing strains *Pseudomonas aeruginosa* PAO1 (6, 7) and *P. stutzeri* A1501 ((8); Figure S8). Psl is reported as a signal to trigger pili-mediated homologous aggregating (9), and Psl homologues were also highly conserved in many *Pseudomonas* species (10). Therefore, the pili-mediated aggregation may be prevalent in *Pseudomonas* species, including our strains.

Figure S8 Polysaccharide synthesis locus (*psl*) in *Pseudomonas* spp. Relative gene organization and protein size are depicted for *Pseudomonas aeruginosa* PAO1 and *P. stutzeri* A1501, and *P. stutzeri* AN10. Information was gathered by analyzing the genome of the three *Pseudomonas* strains (accession number: PAO1, NC_002516.2; A1501, CP000304.1; AN10, CP003677) using NCBI tBlast(https://blast.ncbi.nlm.nih.gov/Blast.cgi?PROGRAM=tblastn&PAGE_TYPE=BlastSearch&LINK_LOC=blasthome).

We attached this analysis as Figure S8 in RM. The related discussion has been revised accordingly in **line 197-208** in RM:

“How type IV pilus contributes to the development of ‘bubble-burst’ pattern? Previous studies indicate that type IV pilus plays essential roles in mediating twitching motility (22), stabilizing interactions between cells and the abiotic surface (23), as well as facilitating cell-to-cell adhesion (24, 25), which is required for aggregate formation. Several studies based on microscopic investigation directly observed that *Pseudomonas* cells harboring Type IV pilus could migrate along the surface and gather together by recruiting adjacent and homologous cells, resulting in

the formation of multicellular aggregates (23). This pilus-mediated aggregating is affected by Psl exopolysaccharide (29), and the key genes involved in Psl synthesis are located in the genome of our strains (Figure S8). Therefore, we hypothesize that the observed ‘bubble’-like structures in our study may be derived from pilus-mediated formation of cell aggregates.”

I have two main scientific criticisms. First, I am not convinced that the intermixing index used in this paper is capturing important nuance. The index is the number of intersections between the two populations divided by the circumference being analyzed. This does not take into account the size of each sector. A major claim in this paper is that the cross-feeding community is less intermixed than the competitive community, based upon this index. However, if I examine the microscopy in Fig. 1, I see that the green sectors are quite thin. Therefore, the green cells are usually much closer to the red cells in the cross-feeding community than is the case for the competitive community. I would argue that from the green cells' perspective, the cross-feeding community is more intermixed beyond the inoculation area than is the competitive community. Even if one disagrees with that interpretation, in any case the patterns are quite clearly different, yet the index analysis says they are the same at far radii, meaning it is not capturing important information. Could a modified metric be created which captures the clear differences between the two colonies?

Response: Thanks for this helpful comment. Here, we defined the intermixing index following our previous study (11), and this method was also widely used by other studies to quantify intermixing level (5, 12). Nevertheless, we acknowledge that no method is perfect, and we agree that this method cannot capture the difference between the two colonies at far radii. Following your suggestion, we performed additional analyses on the average size of each red and green sector along the radii of the colonies.

Figure S3 Analysis of the mean width of the strain sectors along the growing front line of the colonies developed in different scenarios, as a function of radial distance R from inoculation area. Here, we used crossing statistics to determine the average width of clonal strands (i.e., a continuous strand with corrected fluorescence value above 0).

As shown in Figure S3, we found that the sector sizes of patterns formed by the cross-feeding and competitive community are indeed different at far radii. In particular, the average size of the sectors formed by AN0001 (green in Figure 1) is smaller than that of AN0010 (red in Figure 1) in the cross-feeding community. In contrast, the AN0001-sector (green in Figure 1) is much larger than that of AN0010 (red in Figure 1) in the competitive community. **The result suggests that the cross-feeding community is more intermixed at far radii than that of the competitive community from the perspective of strain AN0001, but less intermixed from the perspective of strain AN0010.** Importantly, we also found that, at the inoculation area (e.g., < 60%R), average sizes of both AN0010-sectors and AN0001-sectors in cross-feeding community are smaller than that in competitive community. This finding suggests that **spatial intermixing of cross-feeding pattern is less mixed than the competitive pattern at the inoculation area (e.g., < 60%R).** Accordingly, we revised our description and conclusion in **line 135-139** in RM, as follow,

“However, our analysis of this pattern suggested that the ‘bubble-burst’ pattern formed in the ‘cross-feeding’ scenario is even less mixed than the pattern formed in the ‘competition’ scenario (Figure 1C; $p = 2.7e-4$), **especially at the inoculation area** (Figure 1C; Figure S3), in disagreement with previous observations ((14, 27); Figure 1 B-C)”.

Associated with the improved statistics (see the below response), we believe our conclusion in RM is more robust.

My second criticism is that the statistics are not well-enough explained. What specific values are being compared in the Mann-Whitney test? The intermixing indices at all radii, at just the range-expansion radii, ...? If all radii (or even more than one radius) were used, how was the non-independence of the datapoints controlled for (as one could do, for example, with a mixed-effects model)?

Furthermore, it isn't clear to me that the interpretation is always valid. For example, on lines 148-149, those p-values suggest that the intermixing index is significantly different between the pili mutant cross feeders and previous studies, yet the sentence says "better resembled the ones that developed in previous similar studies," which seems to be arguing for the absence of a significant difference. Please clarify.

Response: Thanks for this important comment. We discussed with several statisticians, and found our original statistical method based on Mann-Whitney U test is indeed not suitable. We thus revised our statistical method used for comparing the intermixing between different patterns. In the RM, we defined a new parameter, ‘ \sum intermixing index’, equaling to the summation of intermixing index along the radius. Then we compare the ‘ \sum intermixing indexes’ among different patterns using unpaired, two-tailed, Student's t-test.

Figure S2 Statistical analysis of the intermixing level of the patterns developed in different scenarios. Here, we used crossing statistics to determine the average width of clonal strands (i.e., a continuous strand with corrected fluorescence value above 0). We defined a parameter, ‘ Σ intermixing index’, equaling to the summation of intermixing index along the radius. Then we compare the ‘ Σ intermixing indexes’ among different patterns using unpaired, two-tailed, Student's t-test. These analyses were performed in Wolfram Mathematica (version 12.4).

As shown in Figure S2, we found that

- (i) “the intermixing levels of the ‘bubble-burst’ pattern are significantly lower than those generated by cross-feeding consortia performing toluene degradation ((14); $p = 3.9e-12$) and denitrification ((27); Figure 1C; Figure S2; $p = 1.4e-11$)”; (**line 123-126**)
- (ii) “‘bubble-burst’ pattern formed in the ‘cross-feeding’ scenario is even less mixed than the pattern formed in the ‘competition’ scenario (Figure 1C; Figure S2; $p = 2.7e-4$)”; (**line 135-139**)
- (iii) “deactivation of the type IV genes encoding pilus caused the ‘bubble-burst’

pattern to disappear and significantly increased the spatial intermixing of the two interacting populations in the developed pattern (Figure S2; $p = 1.9e-7$); (line 135-139)

(iv) “The mixed pattern that formed also better resembled the ones that developed in previous similar studies ($p = 0.89$ compared with the pattern formed by the toluene-degrading community, and $p = 0.24$ compared with that of the denitrification community), and showed higher intermixing compared to the pattern formed by the same community in the ‘competition’ scenario (Figure S2; $p = 1.1e-8$).”; (line 154-165)

Accordingly, the related Methods description were also revised in (line 359-362) of RM:

“To compare the overall intermixing level of different patterns, a parameter, ‘ Σ intermixing index’, were defined. ‘ Σ intermixing index’ equals to the summation of intermixing index of a pattern along the radius. For comparative statistics, unpaired, two-tailed, Student's t-test was performed. These analyses were performed in Wolfram Mathematica (version 12.4).”

We believe our statistics in RM is more reliable.

Other:

The productivity results with the pili led the authors to an intriguing hypothesis: pili mutants interact better with each other than WT strains, possible due to the increased spatial proximity. Put differently, I believe this would lead the authors to hypothesize that the pili mutants have higher fitness than WT. It would be great to have a test of this, by doing an invasion assay of the pili mutant, and seeing if it increases in frequency. Total community biomass, which is what was measured, may correlate to strain fitness, but they are not 1:1.

Response: Thanks for this insightful suggestion. Accordingly, we performed additional experiments to mimic invasion/evolution. We labelled 0010 Δ *pilAB* and 0001 Δ *pilAB* with tagBFP protein, mixed one of them with the two WT strains

(labeled with EGFP and Mcherry) at a low initial frequency (5%), and grew the community on agarose surfaces. However, experimental results did not resemble the expectation. In the developed pattern, we cannot detect the BFP signal, and the ‘bubble-burst’ (‘bubble-jet’) pattern still formed. When we collected the colony biomass, and measured the relative fraction using a microplate reader, we found that the fraction of invasion pili mutants did not increase (decrease from 5% to 2%~3%; Figure R1A), and the total biomass of the colony did not change (Figure R1B), either.

Figure R2 the pili mutants do not possess higher competitive fitness than WT during the invasion. To mimic the invasion of the pili mutants, we mixed 0010ΔpilAB or 0001ΔpilAB with the two WT strains at a low initial frequency (5%), and grew the community on agarose surfaces. The colonies were collected by an inoculation loop from the plate after 120-h's culture. Bacterial cells were resuspended in 100 μL minimum medium and vortexed for 10 min to destruct the biofilms. fluorescence intensities of TagBFP, eGFP, and mCherry were measured to estimate the final frequency of the invasion strains (A). The OD₆₀₀ was measured to estimate the total biomass (B).

This result indicates that although the pili mutants themselves developed a pattern with higher biomass, they are less competitive when co-exist with the wild-type strains. Type IV pilus is reported to mediate many important activities of bacteria (13,

14), so deactivating pili may decrease the competitive fitness of the mutants against the wild type. We acknowledge that at this stage, we cannot reveal the underlying mechanisms. However, we think that our main point in this paper is proposed at an ecological scale, and we mainly declare that our finding offers a potential strategy to engineer spatial patterns of artificial communities. The invasion experiment aims to mimic the fate of the mutations in the wild-type biofilm, which aims to explain the evolutionary outcomes of natural communities. Therefore, we think the experiment is a little bit beyond the goal of this paper. Nevertheless, we still hope we can shed light on the mechanism in the future studies using some novel technologies, such as single-cell based observations. Hope this response can meet your approval.

Line 135: it argues that changing nutrient concentration didn't affect the qualitative appearance of the bubble-jet pattern; please specify that you mean the average # of bubbles, as all other metrics (intermixing index, bubble area, total biomass, and I believe strain dominance) changed. Furthermore, there is no statistical support testing the # of bubbles, and when I look at S2E, it appears that the median number of bubbles increases monotonically until 10mmol, then stays at a plateau. I would argue this is contrary to the claim in the main text.

Response: Thanks for the comments. Here, 'qualitative appearance did not change' means that 'bubbles' structures are still developed in the pattern, and strain 0010 still expanded from the 'bubbles'. We acknowledge here our claims in the original manuscript did not clarify this conclusion well. In the RM (**line 142-145**), we modified it as:

“These investigations indicated that although these factors did have impacts on the detailed morphology of the pattern, alteration of these factors failed to **totally disrupt** the development of the 'bubble-burst' pattern (Figure S4-S5).”

Abstract: should note the general reason why intermixing has been shown to occur (i.e. mutualism). Arguably one could claim that the best-studied spatial pattern isn't intermixing but sectoring of competitive / neutral populations.

Response: Thanks for the comments. We toned down the statement in RM (line 23-24), as follow:

“Spatial intermixing emerging from microbial interaction is one of the best-studied characteristics of spatial patterns.”

Figure 1A is not clear for defining the cross-feeding relationship. Can AN0010 grow during its upstream partial degradation? Or does it require pyruvate / acetyl-CoA to be generated by AN0001? A better description of the specific needs of each partner is needed. I see this is available in the supp, but I think it is critical enough to be in the main text or Fig 1. I also think proof that they cannot grow on their own on salicylate is important.

Response: We are sorry for this terminology confusion. Yes, strain AN0010 cannot grow alone using salicylate as the sole carbon source (Figure R1), and it requires pyruvate released by strain AN0001 for growth in co-culture (1). In our verification experiments (Figure R1; adapted from our former study (1)), both strain AN0010 and strain AN0001 cannot grow in monoculture using salicylate (Figure R1A). However, when paired and in presence of salicylate, the two strains significantly grew (Figure R1B). Therefore, **both AN0010 and AN0001 benefit from the other, and each partner depends on the other for survival and reproduction, which is a typical association of obligate mutualism, as defined in previous articles (2, 3).**

Figure R1 Identification of the strains used in this study. (A) Growth dynamics of the mono-culture of four strains using salicylate (solid line) or catechol (dash line) as the sole

carbon source. Five replicates were performed for each treatment. 10 C-mmol/L salicylate or catechol were used in the culture experiments. (B) Liquid co-culture dynamics of strain AN0010 and strain AN0001 supplying salicylate as the sole carbon source.

To better clarify this terminology, we made following revisions in the RM:

1. We revised Figure 1A to clarify the mutualistic and bilateral interaction between the two strains.

Figure 1A Schematic shows the bilateral mutualistic interaction between strain *P. stutzeri* AN0010 and strain *P. stutzeri* AN0001 during salicylate degradation. Strain AN0010 degrades salicylate into the intermediate catechol, which feeds strain AN0001 as the substrate for further degradation. However, strain AN0010 cannot obtain direct carbon source from salicylate degradation to support its growth. When paired with strain AN0001, AN0001 degrades catechol to pyruvate, which feeds AN0010 with pyruvate to support its growth.

2. We clarified the that these two strains engage in a mutualistic interaction in **line 93-100**, as follow:

“Strain AN0010 degrades salicylate into the intermediate catechol, but cannot grow in monoculture using salicylate (ref; Supplementary Figure 1). Instead, when the two strains are paired, strain AN0001 further degrades catechol to pyruvate and acetyl-CoA (Figure 1A). These small molecules are released to the medium to support the growth of strain AN0010 (26). Therefore, these two strains engage in a mutualistic interaction and act as a bilateral cross-feeding interaction consortium in the presence of salicylate (26).”

3. The term 'Mutualism' is emphasized in Abstract in **line 29**.

Hope these explanations and revisions can address your concern.

Line 68: a bit too strong to say that spatial mixing alleviates antibiotic stress-perhaps add "can," as it is only referencing one instance

Response: Thanks for your suggestion. The revision has been made in RM.

line 76: "genetic surfing" is jargon, please define or rewrite

Response: Thanks for your suggestion. 'Genetic surfing' was changed to term 'Genetic drift at expanding frontiers' in RM, which is more general.

line 80: "metabolic interactions" is unclear; please specify mutualistic / cooperative

Response: Thanks for your suggestion. We use 'cooperative metabolic interactions' in RM.

Line 142: sentence fragment

Response: Sorry for the mistake. In the RM, the sentence has been modified (**line153-154**), as follow:

"We found that the deactivation of flagellar genes did not change the development of the 'bubble-burst pattern'."

Line 178: grammar mistake

Response: Sorry for the mistake. In the RM, the sentence has been modified (**line 193-195**), as follow:

"the resulting community has better robust against varied initial ratios between the two strains."

Fig 2D, E: I cannot distinguish the circles and squares. I would argue final frequency and biomass should be separate graphs.

Response: Thank you for the suggestion. We agree that separating final frequency

and biomass graphs can make the figure clearer. However, we wish to publish this manuscript as an 'Observation' type. The Figure number and size are largely limited in this article type (<https://journals.asm.org/journal/spectrum/article-types>). Although this suggestion is helpful, we cannot add more figure items to the manuscript, and Figure 2 is currently too large to be added with more sub graphs. Therefore, we suggest here we still keep the current style of Figure 2D, E. Hope this explanation can meet your approval.

Lines 100-101: it argues that AN0010 formed bubble-like structures, then AN0001 surrounded these, but I see no temporal analysis besides a single time series in S1 to support this claim.

Response: Sorry for the mistake and unclarity. This conclusion is actually supported by Figure S1A-B. The time-series images in Figure S1A clear show the process that AN0010 formed 'bubble'-like structure. **Importantly, we analyzed the distribution of fluorescence in confocal images. As shown in Figure S1B-C, we found that the GFP (AN0001) signal is highly concentrated around the Mcherry signal (AN0010). This result suggested that cells of AN0001 tended to distribute around AN0010 in the formed pattern (surrounding).** In RM (line 108), we changed the annotation after this claim to clarify that this conclusion is supported by Figure S1A-C.

A

B

C

Figure S1 Characterization of the spatial pattern formed by our synthetic salicylate-degrading community. (A) Images show the colony growth dynamics of the community when supplying salicylate as the sole carbon source. The growth of typical 'bubble' areas is zoomed in. (B) Confocal imaging shows the three-dimensional structure of a typical 'bubble' area. (C) Analysis of the relative fluorescence intensity of the image showed in (B), suggesting the distribution of the two populations in this area.

"jetted" to me implies movement with force; I think it will lead to confusion and the authors should choose a different descriptor.

Response: Thank you for the suggestion. We revised 'jet' to 'burst' in the RM. 'burst' means to break open or apart, especially because of pressure from inside (Wiktionary: <https://en.wiktionary.org/wiki/burst#Verb>), so we use 'burst' to imply that strain 0010

expands from the ‘bubbles’ in passive way driven by the pressure from inside (cell growth). Accordingly, the description in RM was revised in **line 108-110**

“During range expansion, cells of strain AN0010 expanded from these ‘bubble’ structures, similar to the bubble ‘burst’ into the expanding sectors.”

We believe this term is more suitable.

References cited in this response-to-editor letter:

1. Wang M, Chen X, Tang Y-Q, Nie Y, Wu X-L. 2021. Substrate traits shape the structure of microbial community engaged in metabolic division of labor. bioRxiv doi:10.1101/2020.11.18.387787:2020.11.18.387787.
2. Little AE, Robinson CJ, Peterson SB, Raffa KF, Handelsman J. 2008. Rules of engagement: interspecies interactions that regulate microbial communities. *Annu Rev Microbiol* 62:375-401.
3. Faust K, Raes J. 2012. Microbial interactions: from networks to models. *Nature Reviews Microbiology* 10:538-550.
4. Muller MJI, Neugeboren BI, Nelson DR, Murray AW. 2014. Genetic drift opposes mutualism during spatial population expansion. *Proceedings of the National Academy of Sciences of the United States of America* 111:1037-1042.
5. Momeni B, Brileya KA, Fields MW, Shou WY. 2013. Strong inter-population cooperation leads to partner intermixing in microbial communities. *Elife* 2.
6. Ma LY, Lu HP, Sprinkle A, Parsek MR, Wozniak DJ. 2007. *Pseudomonas aeruginosa* PSI is a galactose- and mannose-rich exopolysaccharide. *Journal of Bacteriology* 189:8353-8356.
7. Jackson KD, Starkey M, Kremer S, Parsek MR, Wozniak DJ. 2004. Identification of psl, a locus encoding a potential exopolysaccharide that is essential for *Pseudomonas aeruginosa* PAO1 biofilm formation. *Journal of Bacteriology* 186:4466-4475.
8. Shang L, Yan Y, Zhan Y, Ke X, Shao Y, Liu Y, Yang H, Wang S, Dai S, Lu J,

- Yan N, Yang Z, Lu W, Liu Z, Chen S, Elmerich C, Lin M. 2021. A regulatory network involving Rpo, Gac and Rsm for nitrogen-fixing biofilm formation by *Pseudomonas stutzeri*. *NPJ Biofilms Microbiomes* 7:54.
9. Zhao K, Tseng BS, Beckerman B, Jin F, Gibiansky ML, Harrison JJ, Luijten E, Parsek MR, Wong GCL. 2013. Psl trails guide exploration and microcolony formation in *Pseudomonas aeruginosa* biofilms. *Nature* 497:388-+.
 10. Mann EE, Wozniak DJ. 2012. *Pseudomonas* biofilm matrix composition and niche biology. *FEMS Microbiol Rev* 36:893-916.
 11. Goldschmidt F, Regoes RR, Johnson DR. 2017. Successive range expansion promotes diversity and accelerates evolution in spatially structured microbial populations. *Isme Journal* 11:2112-2123.
 12. Pielou EC. 1966. Species-diversity and pattern-diversity in the study of ecological succession. *J Theor Biol* 10:370-83.
 13. Craig L, Forest KT, Maier B. 2019. Type IV pili: dynamics, biophysics and functional consequences. *Nature Reviews Microbiology* 17:429-440.
 14. Maier B, Wong GCL. 2015. How Bacteria Use Type IV Pili Machinery on Surfaces. *Trends in Microbiology* 23:775-788.

Finally, we greatly appreciate the helpful comments from the reviewers and the editor, and hope that our modifications and responses adequately address the concerns raised. All of these modifications can be tracked in the marked-up manuscript.

January 23, 2022

Prof. Xiao-Lei Wu
College of Engineering, Peking University
Beijing
China

Re: Spectrum01944-21R1 (Type IV pilus shapes a 'bubble-jet' pattern opposing spatial intermixing of two interacting bacterial populations)

Dear Prof. Xiao-Lei Wu:

Thank you for submitting your manuscript to Microbiology Spectrum. The reviewers appreciated your revisions and it has made the manuscript much stronger. There are a few more minor things that I would like you to address before we can accept your manuscript for publication. These minor issues are noted in the reviewer comments at the bottom of this email.

Link Not Available

Sincerely,

Benjamin Wolfe

Journals Department
Reviewer comments:

Reviewer #2 (Comments for the Author):

In my opinion, the authors have adequately addressed the reviewers' comments. I only have one minor suggestion:

If I understand correctly, the authors are arguing that neither AN0010 nor AN0001 grow on salicylate as the only carbon source, whereas the coculture grows on salicylate (under certain conditions). In other words, when in an environment with salicylate, both species grow better compared to how they would grow on their own. I think this is sufficient evidence, but it needs to be spelled out to make it clear to your readers. The addition of Fig 1A is certainly helpful, but in my opinion, the description in the rebuttal letter is clearer compared to what is included in lines 93-100 of the manuscript.

Reviewer #3 (Comments for the Author):

Thank you to the authors for their revisions; the altered statistics, and the restructuring of the intro, as well as the improved Fig 1

model have all helped to improve the manuscript.

Thank you for the new intermixing metric, I think it adds some clarity. Could a sentence be added to the MS that describes the interesting fact that each of the cross-feeding strains experiences a different amount of intermixing? This asymmetry seems interesting and worth highlighting.

I think that some of the previous suggestions would be nice to have in this version (e.g. separation of freq / biomass into two panels), but understand that the authors are feeling constrained by the format.

I think it would be useful to put the type of statistical test used, and the number of replicates, into the main text, at least the first time the stats are introduced (e.g. line 123), so that readers without the supp are not wondering where the p-value originates from.

I wonder if the authors could add one or two sentences to the conclusion for what their results might mean when we think about cross-feeding in natural communities. Should we use the presence of T4 pili as an indicator that the species is probably not an obligate crossfeeder?

Finally, some grammar mistakes and typos remain, for example this one: "the resulting community has better robust against varied initial ratios between the two strains" should be "the resulting community was more robust to variable initial ratios..."

Staff Comments:

Preparing Revision Guidelines

Please return the manuscript within 60 days; if you cannot complete the modification within this time period, please contact me. If you do not wish to modify the manuscript and prefer to submit it to another journal, please notify me of your decision immediately so that the manuscript may be formally withdrawn from consideration by Microbiology Spectrum.

Responses to the Editor's comments

Thank you for submitting your manuscript to Microbiology Spectrum. The reviewers appreciated your revisions and it has made the manuscript much stronger. There are a few more minor things that I would like you to address before we can accept your manuscript for publication. These minor issues are noted in the reviewer comments at the bottom of this email.

Reponses: We sincerely thank you and the reviewers for the second round of reviewing of our manuscript, and providing additional comments that is very helpful to our manuscript. We have carefully revised our manuscript accordingly to address the issues noted by the reviewers. We hope that you will find this new version of our manuscript improved much.

Responses to the Reviewer's comments

Reviewer #2 (Comments for the Author):

In my opinion, the authors have adequately addressed the reviewers' comments. I only have one minor suggestion:

If I understand correctly, the authors are arguing that neither AN0010 nor AN0001 grow on salicylate as the only carbon source, whereas the coculture grows on salicylate (under certain conditions). In other words, when in an environment with salicylate, both species grow better compared to how they would grow on their own. I think this is sufficient evidence, but it needs to be spelled out to make it clear to your readers. The addition of Fig 1A is certainly helpful, but in my opinion, the description in the rebuttal letter is clearer compared to what is included in lines 93-100 of the manuscript.

Response: We greatly thank you for your positive comments on our paper.

We are very sorry for not clarify this point in the manuscript. In the updated version, we benefited from the previous rebuttal letter to rewrite this part in the manuscript, in line 84-94 (the marked version) as follow:

“In our previous work, we found that that either strain AN0010 or strain AN0001

could not grow alone using salicylate as the sole carbon source (26). When paired and in the presence of salicylate, the two strains grew and acted as a cross-feeding consortium: strain AN0010 degrades salicylate into the intermediate catechol, and strain AN0001 further degrades catechol to pyruvate and acetyl-CoA (Figure 1A). These small molecules can be released to the medium to support the growth of strain AN0010 (26). Therefore, either strain depends on the other for survival and reproduction in the presence of salicylate, so that the two strains engage in a mutualistic interaction and act as a bilateral cross-feeding consortium (Figure 1A).” We believe now the description becomes clearer to the readers.

Reviewer #3 (Comments for the Author):

Thank you to the authors for their revisions; the altered statistics, and the restructuring of the intro, as well as the improved Fig 1 model have all helped to improve the manuscript.

Response: Many thanks for your positive evaluation of our paper, and insightful additional comments helping us improve the manuscript. We have tried our best to address your newly noted issues. **The page and line numbers in the marked version of the revised manuscript (RM) where changes have been made were indicated in each response.** Hope these revisions can meet your approval.

Thank you for the new intermixing metric, I think it adds some clarity. Could a sentence be added to the MS that describes the interesting fact that each of the cross-feeding strains experiences a different amount of intermixing? This asymmetry seems interesting and worth highlighting.

Response: Thanks for this helpful suggestion. We added one sentence to highlight this point in line 131-134 of RM, as follow:

“In the ‘cross-feeding’ scenario, the average size of the sector formed by AN0010 is larger than that of by AN0001 at far radii, suggesting that the strain may possess higher fitness during range expansion in this scenario (Figure S3).”

I think that some of the previous suggestions would be nice to have in this version (e.g. separation of freq / biomass into two panels), but understand that the authors are feeling constrained by the format.

Response: Thanks for this suggestion again. We have tried several format revisions following you and other reviewers' comments, but the initial-check editor declined these revisions because of the format limitation of the 'Observation' type. In particular, we are very sorry that we cannot add more figure items or enlarge the current figures (For example, separation of freq / biomass into two panels in Figure 2). We are very sorry about these settings but we think they may not bother the main flow of the manuscript. Hope this explanation can meet your approval.

I think it would be useful to put the type of statistical test used, and the number of replicates, into the main text, at least the first time the stats are introduced (e.g. line 123), so that readers without the supp are not wondering where the p-value originates from.

Response: Thanks for this important suggestion. We mentioned our statistical method when it was first introduced, in line 117-118 of RM, as follow:

“We found that the intermixing levels of the ‘bubble-burst’ pattern are significantly lower than those generated by cross-feeding consortia performing toluene degradation ((14); **unpaired, two-tailed, Student's t-test**: $p = 3.9e-12$) and denitrification ((27); Figure 1C; Figure S2; $p = 1.4e-11$).”

I wonder if the authors could add one or two sentences to the conclusion for what their results might mean when we think about cross-feeding in natural communities. Should we use the presence of T4 pili as an indicator that the species is probably not an obligate crossfeeder?

Response: Thanks for this helpful suggestion. We revised one of our conclusion sentences to specify how our results on may help understand cross-feeding interactions in natural communities, in line 202-206 of RM, as follow:

“Our results suggest that the presence of cell constituents, such as type IV pili, may

oppose metabolic interactions between different genotypes in surface-attached communities, which further up-scales to influence community-level properties.”

Finally, some grammar mistakes and typos remain, for example this one: "the resulting community has better robust against varied initial ratios between the two strains" should be "the resulting community was more robust to variable initial ratios..."

Response: We are very sorry for the grammar issues. We have revised the mentioned mistakes. We also went through the manuscript several times to improve our language use.

Finally, we greatly appreciate the helpful comments from the reviewers and the editor and hope that our modifications and responses adequately address the concerns raised. All of these modifications can be tracked in the marked-up manuscript.

January 28, 2022

Prof. Xiao-Lei Wu
College of Engineering, Peking University
Beijing
China

Re: Spectrum01944-21R2 (Type IV pilus shapes a 'bubble-jet' pattern opposing spatial intermixing of two interacting bacterial populations)

Dear Prof. Xiao-Lei Wu:

I am pleased to let you know that your manuscript has been accepted for publication in Microbiology Spectrum. I am forwarding it to the ASM Journals Department for publication. You will be notified when your proofs are ready to be viewed.

Sincerely,

Benjamin Wolfe
Editor, Microbiology Spectrum

Journals Department
Supplemental file1: Accept